# Do Not Trust What They Tell: Exposing Malicious Accomplices in Tor via Anomalous Circuit Detection

## ABSTRACT

The Tor network, while offering anonymity through traffic routing across volunteer-operated nodes, remains vulnerable to attacks that aim to deanonymize users by correlating traffic patterns between colluded Entry and Exit nodes in circuits. This paper presents a novel approach for detecting anomalous circuits in the Tor network, and for the first time provides a more comprehensive identification of potential malicious accomplice nodes in Tor by taking roles of nodes in anomalous circuits into consideration. Our method strategically utilizes modified Middle nodes to capture traffic data, followed by a novel circuit classification based on traffic patterns to pinpoint concerned circuits. Two kinds of anomalies are identified: routing anomalies and usage anomalies, that respectively represent the anomalies with explicit or implicit violation of Tor's circuit construction guidelines. This leads to a successful revealing of totally 1,960 anomalous nodes in Tor. Furthermore, we apply clustering analysis with considering corresponding anomalous circuits and other key characteristics to the detected anomalous nodes, revealing potential hidden organizations behind these nodes that can threaten the network's security. Our findings highlight the necessity for the Tor project to adopt targeted mitigation strategies to enhance overall network security and privacy.

## CCS CONCEPTS

• **Security and privacy** → **Pseudonymity, anonymity and untraceability**; **Privacy protections**; **Security protocols**; **Distributed systems security**.

## KEYWORDS

Tor network, Anonymity, Anomalous circuit, Traffic analysis

## 1 INTRODUCTION

The Onion Router (Tor) [1] is a decentralized overlay network that employs multi-layer encryption and multi-hop routing to establish secure and anonymous communication channels. Tor supports a variety of anonymous TCP services, including web servers, email, and SSH. The network relies on nodes and bandwidth resources contributed by volunteers across the globe, which prevents any single entity from exerting complete control over the infrastructure. This makes Tor a vital tool for enhancing privacy on the web, positioning it as an essential component of web security. Despite its strengths, Tor remains vulnerable to malicious nodes deployed as Sybils [2][3][4]. These nodes attempt to be selected by users as Entries and Exits within the same circuit, aiming to perform traffic correlation attacks that could compromise user identities [5][6][7]. Since these attacks present a significant threat to the privacy and security of users browsing with Tor, it is crucial to detect such malicious accomplices and mitigate risks to Tor and web security. Here, a circuit is a communication path consisting of multiple nodes between the sender and receiver that can be divided into several sections. The notation "Sybil" represents a single entity that creates multiple identities to gain disproportionate influence over the network, thereby compromising the security and integrity of Tor. It has been demonstrated that such an approach can identify and trace users' communications, despite Tor's robust multi-layer encryption and routing mechanisms.

Although Tor implements rigorous automatic circuit construction rules to mitigate de-anonymization vulnerabilities, the outcome still contains deficiencies. To prevent de-anonymity attacks, Tor's official guidelines for automatic circuit construction stipulate that two nodes violating certain criteria—such as belonging to the same family group or being within the same /16 subnet—cannot be part of the same circuit [8]. However, In addition to the aforementioned automatic construction, clients in Tor are also allowed to manually select nodes on some of their circuits by modifying the configuration file. These rise vital issues regarding circuit construction in two-folds: On the first fold, users manually building circuits may use their selected/preferred nodes that directly violate Tor's circuit construction restrictions; On the second fold, circuits built according to Tor's rules still may not be secure. Previous works [9][10][11][12] have shown that nodes in Tor can intentionally conceal their family belongings. We further reveal in this work that if these nodes play as Entries and Exits in circuits, Tor's circuit construction regulations can be circumvented and Sybil de-anonymity attacks can be carried out. Later experimental statistics in this work reveal that such phenomena in both folds already evidently exist in the currently operating Tor network, posing a severe and yet overlooking threat to the anonymity of Tor.

Consequently, analyzing node associations in Tor, especially exposing both explicit and implicit node associations (such as, the de facto belonging to the same family) in circuits, is crucially important for preventing Sybil attacks and protecting the anonymity and privacy of Tor. Although previous work [9] demonstrates the potential of concealing node family belongings by comparative study of node configuration properties, it does not consider anomalous circuits and their roles in exposing node associations. **To the best of our knowledge, we introduce the first approach for detecting anomalous circuits and thereafter investigating the organizations behind the involved nodes in this work.** Our approach could reveal implicit node associations that are otherwise impossible to be detected solely via node configuration comparisons and is validated by conducting a large-scale experiment in real-world Tor, offering an indispensable means of exposing anomalous circuits and implicit node families/organizations, thereby enhancing the protection of Tor's anonymity.

First and foremost, our approach focuses on detecting two kinds of anomalous circuits in Tor: (1) when users deliberately select specific node pairs during circuit construction, particularly Entry-Exit pairs, and explicitly disobey Tor's regulations as represented by

Tor's own circuit construction rules; (2) when certain Entry-Exit pairs (self-claimed as not being in the same family) in circuits are used at unusually high frequencies, significantly deviating from the expected probability, which as well strongly suggests the artificial selection of these nodes and the potentially concealed family/organization associations between them. We refer these two kinds of anomalous circuits as Routing anomalies and Usage anomalies respectively. To accomplish the detection, we deploy our own Middle nodes to capture traffic data in circuit sections. We further develop a circuit classification algorithm based on Tor's protocol layer traffic to determine the type of circuits, enabling us to analyze both the purpose of each circuit and the position of nodes within circuits. These techniques help us to identify the concerned Exit circuit where anomalous circuits can possibly happen and obtain Entry-Exit pairs for validation. After detecting anomalous circuits and obtaining anomalous Entry-Exit node pairs, we further investigate whether users' preferences for these node pairs reveal any organizational patterns behind them via clustering algorithm that for the first time takes anomalous circuits into consideration, and comprehensively identify potential Sybils capable of compromising the anonymity of the Tor network.

To sum up, we make the following key contributions:

- We are the *first* to propose an approach for identifying circuits with Usage anomalies, primarily by developing a novel statistical model to detect Entry-Exit pairs with unusual high occurrence frequencies. It contributes to the total detection of 1,960 anomalous nodes in experiments. .
- We propose a technique to determine the position of a controlled Middle node within a specific type of circuit by classifying the circuit section that includes the node. Through validation, the accuracy of the circuit section classification reaches 100%. Such technique serves as a preliminary for our anomaly circuit detection approach and provides informative circuit classification knowledge in Tor.
- We uncover potentially concealed organizational relationships among nodes involved in anomalous Entry-Exit pairs. For the *first* time, roles of nodes in anomalous circuits are considered in the distance measure of clustering. Ultimately, we expose organizational connections among several family groups and discrete nodes within the Tor network.

## 2 BACKGROUND

In this section, we briefly introduce the Tor network and its circuits.

### 2.1 Tor network

The Tor network is a decentralized system designed to provide anonymity and privacy by routing internet traffic through a series of volunteer-operated servers called nodes (or relays). The core principle of Tor involves encrypting data typically three times and then routing it through several randomly selected nodes. Each node is responsible for either peeling off one layer of encryption or adding one, depending on the direction of the communication. The fundamental unit of communication within the Tor network is the Tor Cell. Tor typically uses fixed-length Tor Cell packets, each 514 bytes in size. The basic structure of these packets is outlined in Figure 1.

Here are some terms related to the Tor network.

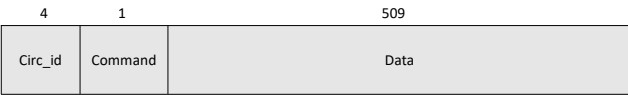

**Figure 1: Tor Cell format**

- **Client**: The Tor client is the user software for connecting to and utilizing the Tor network.
- **Hidden Services(HS)**: HS refer to websites or services hosted within the Tor network. Hidden services have ".onion" domains, which can only be accessed through browsers that support the Tor network.
- **Hidden Service Directory(HSDIR)**: HSDIR stores and provides clients with introduction points, public keys, and other information for hidden servers.
- **Introduction Point(IPO)**: IPO in the Tor network serves as an intermediary between the client and the hidden service.
- **Rendezvous Point(RPO)**: RPO acts as an intermediary that allows clients and hidden services to connect without revealing their IP addresses to each other.

Tor enables anonymous access to both general websites and hidden services by building different kinds of circuits: Exit circuits and Internal circuits. Exit circuits, which include Exit nodes, are primarily used for users to anonymously access normal websites. Three relay nodes are selected to construct a general Exit circuit for communication, serving as the Entry node, Middle node, and Exit node, respectively. Figure 2 illustrates the Cells exchanged between the client and the website service in a general Exit circuit.

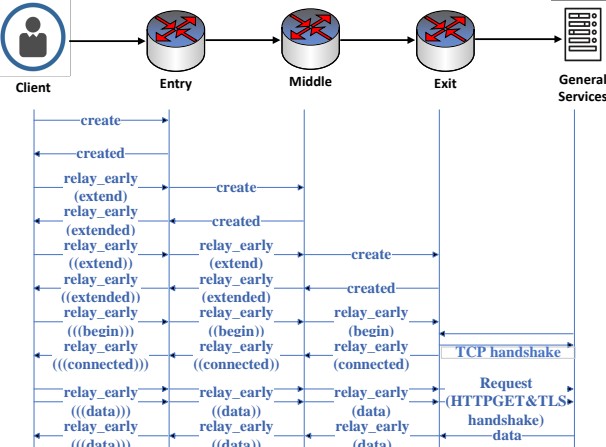

**Figure 2: Exit TOR circuit with exchanged Cells**

In contrast, internal circuits do not contain Exit nodes and are mainly used for the publication and access of hidden services, as well as for maintaining Tor relay information. When the Tor hidden server starts, it selects three Entry nodes as its front proxies and uploads their introduction points and public key information to the hidden service directory server. Clients access this information by creating a three-hop circuit to the HSDIR. The client then selects a rendezvous point and informs the hidden server through the IPO. Both the client and hidden server establish circuits to the RPO, forming a six-hop link for communication. This setup ensures that no single relay can learn the IP addresses of both the client and the

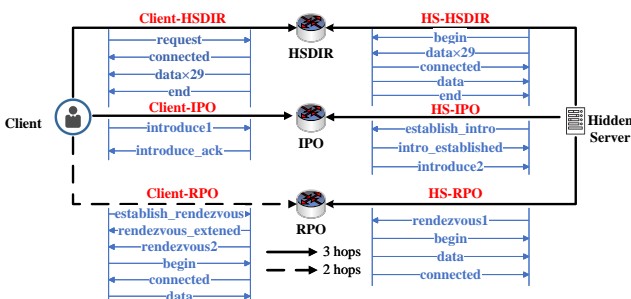

**Figure 3: Internal Tor circuit with exchanged Cells in dark web browsing**

hidden server, maintaining their anonymity. Figure 3 illustrates the Cells exchanged in internal circuits after path construction.

To enhance performance and security, Tor allows data to be sent immediately after a begin Cell without waiting for a connected Cell response, thereby reducing latency and improving transmission efficiency. Additionally, Tor obscures circuit characteristics by adding padding to Cell traces, making traffic analysis more difficult.

## 2.2 Tor circuit construction

Tor clients and hidden servers begin building multi-hop circuits as soon as they have enough directory information and use a bandwidth-weighted routing algorithm[13] to select circuit nodes. In a circuit, the nodes connected to a client or a hidden server are referred to as **Entry** nodes, the nodes that exit the Tor network to connect to the external internet are known as **Exit** nodes, and all other nodes are called **Middle** nodes.

Tor nodes have different capabilities and can be classified into three categories: nodes with the "Exit" flag (denoted as set $E$), which serve as Exit nodes; nodes with the "Guard" flag (set $G$), which can either be Entry or Middle nodes; and nodes with neither "Exit" nor "Guard" flags (set $M$), which are restricted to be Middle nodes only. Notably, sets $G$ and $E$ have a non-empty intersection (i.e. some nodes contain both the "Guard" and "Exit" flags and can serve as either Entry or Exit nodes), while $M$ has no intersection with either $G$ or $E$. The Tor network periodically samples and measures the bandwidth of its nodes, and combines this with their uptime and stability to calculate/update the consensus weights. For example, if we define C[j] to be the consensus weight of nodes j, then the probability of choosing nodes j ∈ E as the Exit node in a circuit can be calculated by Equation (1). Entry nodes are chosen in the analogous way.

$$w^e[j] = \frac{C[j]}{\sum_{j' \in E} C[j']} \tag{1}$$

The Middle node can be chosen from both G and M, the probability of a node chosen for the Middle node can be calculated by equation (2):

$$w^m[j] = \frac{W_{mg}C[j]}{\sum_{j' \in G} W_{mg}C[j'] + \sum_{j' \in M} C[j']} \tag{2}$$

where $W_{mg}$ indicates a multiplier to balance bandwidth among G and M which can be computed as equation (3):

$$W_{mg} = \begin{cases} \frac{\sum_{j' \in G} C[j'] - \sum_{j' \in M} C[j']}{2\sum_{j' \in G} C[j']} & \text{if } j \in G \\ 1 & \text{if } j \in M \end{cases} \tag{3}$$

The types of circuits in Tor include Exit circuits, as shown in Figure 2, and seven types of internal circuits. Six of which—namely *Client-HSDIR, Client-IPO, Client-RPO, Client-HSDIR, HS-HSDIR, HS-IPO*, and *HS-RPO*—are depicted in Figure 3 (as each directed arrow respectively, each "hop" represents a node in the circuit.), along with a *network status circuit*, which is used to obtain network status information. Additionally, there is a special category of circuit, known as *prebuilt circuits*, which are constructed in advance to quickly respond to new requests of building an Exit or internal circuit. In summary, there are totally 9 types of circuits in the Tor network.

## 3 PROBLEM STATEMENT

In this section, we introduce the threat model and the main challenges we face, followed by the basic idea.

### 3.1 Threat model

We assume attackers obey Tor's operating rules except that they conceal the association among their own deployed nodes, attempting to de-anonymize user activities in Tor. As shown in Figure 4, the attacker deploys a set of nodes within the Tor network, pretending that these nodes are independent of each other. The attacker, along with trusted users who donate nodes to the attacker, tend to select these trusted nodes when using Tor, particularly as designated Entry and Exit nodes, in an effort to protect their own communication anonymity. If a regular user's client unintentionally selects these attacker-controlled nodes as Entry-Exit pairs in an Exit circuit. The attacker can correlate traffic and compromise user anonymity. Using the method in this paper, we deployed our Middle nodes to capture and detect these fixed Entry-Exit pairs.

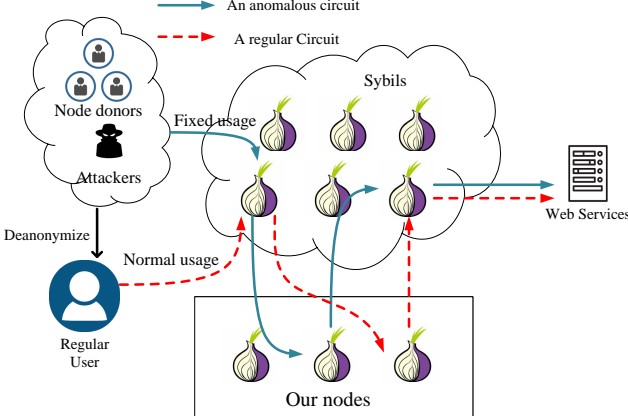

**Figure 4: Threat model**

### 3.2 Basic idea

Our goal is to detect anomalous circuits with Entry-Exit node pairs chosen by users that may have explicitly or implicitly violated Tor's circuit construction guidelines. Further, we mine potential sybils from these involved Entry-Exit pairs. Since only in **Exit circuit** that the client has the ability to fix both endpoints (Entry-Exit nodes) of the circuit, we intend to firstly identify Exit circuits and collect corresponding node information for validation. Also, the prebuilt circuits include a subset designated for future use in Exit circuits which can also be utilized for validation.

We start by deploying our own Middle nodes within the real Tor network, which will be selected by various types of circuits. By modifying the node's source code, we log Cell traces and adjacent node information within circuit sections. Subsequently, utilizing traffic features from Tor's protocol layer, we design a circuit section classification method based on the totally 24 potential circuit section types (each type of circuit consists of two or three circuit sections that involve a Middle node, see Appendix A). In this way, information of Entry-Exit pairs in Exit circuits can be discerned and collected. To carry on, as aforementioned, we define two types of circuit anomalies: **Routing anomalies**, where circuits explicitly violate Tor's routing rules, clearly indicating that the corresponding Entry-Exit pairs are artificially selected; and **Usage anomalies**, where circuits has node pairs with unusual high occurrence rates, highly indicating artificially fixed Entry-Exit pairs. Finally, we propose a clustering analysis based on three features: similarity, anomalous connectivity, and family group flag, to investigate whether users' preferences for these node pairs exhibit any organizational patterns, thereby enabling the identification of Sybils in Tor.

Identifying an anomalous Entry-Exit pair in the Tor network involves two significant challenges as follows: (1)**Discerning the Exit circuits**. We need to accurately isolate the Exit circuits from all other circuits captured by the traffic logs of our Middle node. This is complicated due to several factors including the variety of circuit types in Tor, the fixed-length nature of communication Cells, and the use of padding mechanisms designed to obscure specific traffic patterns; (2)**Building a statistical model to detect unusual Entry-Exit pair frequencies.** This task involves distinguishing between "fixed Entry-Exit pairs" and cases where only a "single fixed Entry or Exit" is involved. Additionally, the model must account for the natural fluctuations in Tor's consensus weights over time, which influence the selection probabilities of nodes.

## 4 METHODOLOGY

This section provides a detailed description of our approach. We begin by determining node positions, detecting anomalous circuits, and clustering anomalous nodes to uncover hidden Sybils.

### 4.1 Identifying node position in a circuit

In order to identify Exit circuits via classification for the next step of anomaly detection, our first step involves modifying the source code of our Tor node to record Tor traffic, including Cell sequences with directions, timestamps and circuit section ID. A circuit section ID along with its Cell sequences is referred to as a Cell trace, which forms the basis of our dataset. The regular Tor traffic could generate a total of 24 types of circuit sections, deriving from all 9 types of circuits. In addition to these circuit types, there may be other puculiar circuits that arise from user's unofficial modifications to Tor, which we do not consider in our analysis.

To begin with, we need to eliminate noises from the Cell traces. This involves padding Cells used in Client-IPO and Client-RPO circuits to obscure circuit characteristics. These padding Cells must be identified and removed to prevent circuit recognition disruptions. Since the drop command in a padding Cell is encapsulated within a relay Cell, nodes cannot read it directly, we distinguish these padding Cells based on their timestamp characteristics instead.

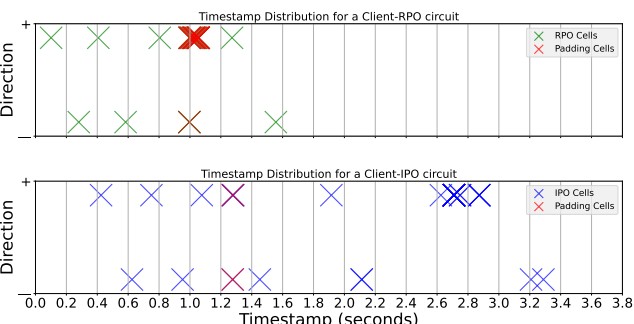

**Figure 5: Cell trace of the Entry-Middle section within Client-IPO and Client-RPO circuits**

Figure 5 shows examples of the real ClientIPO and ClientRPO circuits for which we created and collected data. The "+" direction indicates traffic sent from the circuit's starting point to the destination, while the "−" direction indicates the opposite. Padding Cells only occur in the Entry-Middle section. In the Client-IPO circuit, padding Cells are added starting at the sixth Cell, beginning with a Cell in the "+" direction, followed by approximately 10 Cells in the "−" direction. In the Client-RPO circuit, padding Cells are added starting at the sixth Cell, formatted as one or more Cell pairs with directions alternating between "+" and "−".

We apply a DBSCAN (Density-Based Spatial Clustering of Applications with Noise) [14] clustering algorithm to isolate timestamp-dense padding Cells from a Tor Cell trace. DBSCAN is commonly used to handle data with noise or outliers. It operates by identifying core points that have a sufficient number of neighbors (determined by min_samples) within a specified radius (eps). Points that are densely packed within this radius are grouped into clusters, other are considered as noise. We set eps to 0.1s and min_samples to 2 in our setup. Specifically, If a Cell trace begins forming a dense region starting from the sixth Cell, and the direction follows the pattern of Client-RPO or Client-IPO padding, the circuit is labeled with ifpadding = true and the dense region is removed. By removing these padding Cells, we can obtain the original Client-IPO and Client-RPO Cell traces and begin studying the traffic features of each circuit. Figure 6 describes the decision tree used for circuit section type classification. The classification encompasses all 24 types, which are thoroughly outlined in Appendix A.

We speculate that every circuit section has its distinct Cell trace characteristics that can distinguish them from other. After carefully inspecting features of each circuit type and utilize integrated feature representation considering critical features including both the direction of each Cell and the sequential directional features of the overall Cell trace, we successfully establish the classification decision tree in Figure 6. It is worth mentioning that there exist circuits sections that cannot be easily distinguished directly from their Cell trace characteristics. For example, Client-IPO circuits have an equal number of incoming and outgoing Cells, overlapping with some sections of prebuilt circuits. Some of these positions cannot be directly distinguished based on their features. Therefore, in addition to traffic sequences, we innovatively utilize the "ifpadding" label to help distinguish these circuits.

In order to verify the accuracy of the circuit features, we design the following experiment: we set up clients and hidden services (HS)

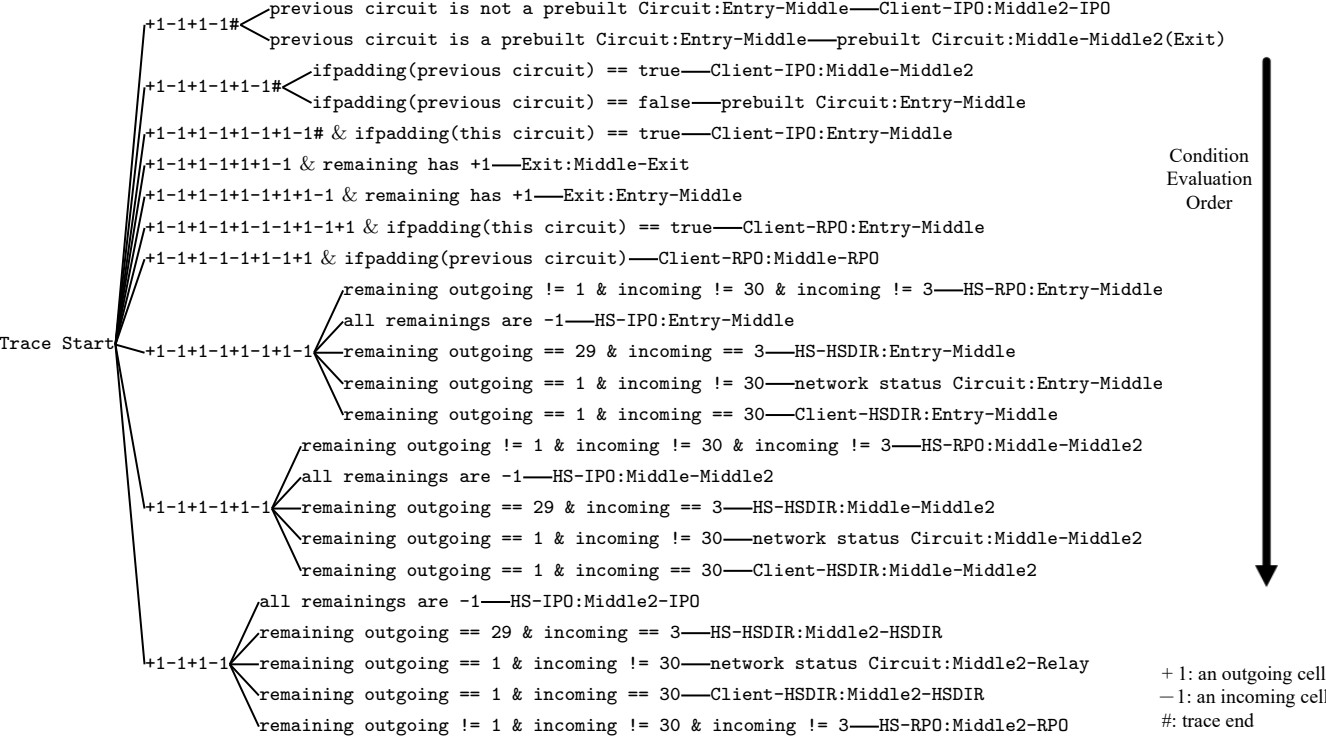

**Figure 6: Decision tree for classifying circuit sections**

in the real Tor network, and modify the source code of client and HS so that all Middle nodes are selected from the nodes we deploy. Then, we visit 100 open sites and 100 hidden service sites respectively, and print out the information of each circuit (fingerprint of each hop in circuit). During each path construction, the corresponding Cell trace of the selected Middle node is recorded (associated by timestamp and circuit section ID), and the actual Cell trace is compared with the theoretical analysis in section before. All the experiment produced 100 Exit circuits, and 600 internal circuits (100 each for 6 types), and 100 prebuilt circuits. The results demonstrate that our padding-removal method effectively eliminates padding in specific circuits, and the classification of circuit sections remains at **100%** accuracy.

## 4.2 Detecting anomalous circuit

Based on previous circuit classification, if our controlled node is identified as the second hop in an Exit circuit, or in a prebuilt circuit where the third hop has an "Exit" flag but not a "Guard" flag(Such nodes can only be used as Exit nodes), we can conclude that our own node is involved in Exit activities. Corresponding circuits contains all Exit circuits and a subset of prebuilt circuits such that the fingerprints of both Entry and Exit nodes can be obtained. We utilize the unique fingerprints of the Entry-Exit pair to conduct anomalous circuit detection. Among the two types of anomalous circuits mentioned, Routing anomalous circuits can be easily identified through simple configuration comparisons. Therefore, our primary focus will be on detecting Usage anomalous circuits.

Assume the probability of selecting node $i$ as the Entry is $P_{\text{Entry}}(i)$, and the probability of selecting node $j$ as the Exit is $P_{\text{Exit}}(j)$. Then, the conditional probabilities $P(i_{\text{Entry}} \mid j_{\text{Exit}})$ and $P(j_{\text{Exit}} \mid i_{\text{Entry}})$,

which represent the likelihood of selecting node $j$ as the Exit given that node $i$ is the Entry, and vice versa, should be equal to $P_{\text{Entry}}(i)$ and $P_{\text{Exit}}(j)$ respectively, given the anticipated independence of such events. Assume that there are $N$ unique Entry-Exit pairs gathered during the time span of data collection. Among these, the number of circuits with node $i$ as the Entry is denoted by $N_{\text{Entry}}^i$, and the number of circuits with node $j$ as the Exit is $N_{\text{Exit}}^j$. The number of circuits where node $i$ and node $j$ simultaneously serve as the Entry and Exit nodes, is denoted by $N_{ij}$. Then, $F(i_{\text{Entry}} \mid j_{\text{Exit}})$ and $F(j_{\text{Exit}} \mid i_{\text{Entry}})$ can be formally expressed by Equation (4).

$$F(j_{\text{Exit}} \mid i_{\text{Entry}}) = \frac{N_{ij}}{N_{\text{Entry}}^i}$$

$$F(i_{\text{Entry}} \mid j_{\text{Exit}}) = \frac{N_{ij}}{N_{\text{Exit}}^j} \tag{4}$$

Thus, $F(i_{\text{Entry}} \mid j_{\text{Exit}})$ and $F(j_{\text{Exit}} \mid i_{\text{Entry}})$ represent the frequencies of the probabilities $P(i_{\text{Entry}} \mid j_{\text{Exit}})$ and $P(j_{\text{Exit}} \mid i_{\text{Entry}})$. If the independence of node selection holds during determining every Entry-Exit pair, we expect that $F(i_{\text{Entry}} \mid j_{\text{Exit}})$ and $F(j_{\text{Exit}} \mid i_{\text{Entry}})$ shall approach $P_{\text{Entry}}(i)$ and $P_{\text{Exit}}(j)$ respectively, when $N \rightarrow \infty$.

However, a great challenge is that the practical collection time span of all Entry-Exit pairs and the reflected conditional frequencies are with respect to a duration of **more than forty days**. Although the probabilities of $P_{\text{Entry}}(i)$ and $P_{\text{Exit}}(j)$ at any moment can be calculated using the equations presented in Section 2.2, however, due to fluctuations in the total number of nodes, the global consensus weight continually varies, causing these probabilities to fluctuate frequently during the time span. To address such a challenge, we draw on Quastel's binomial model for quantal neurotransmitter

release [15], which accounts for uncertainty stemming from fluctuating probabilities. The author propose that if each individual probability of the success event $P_{\text{success}}$ within the fluctuations remains below a threshold of 0.3, the event can be effectively modeled by a simple binomial distribution [16] with an adjusted success probability. In our case, given $P_{\text{Exit}}(j)$ as an example, the adjusted success probability $P'_{\text{Exit}}(j)$ is then given by equation (5), where $\text{Var}(P_{\text{Exit}}(j))$ represents the variance of $P_{\text{Exit}}(j)$ and $E(P_{\text{Exit}}(j))$ represents its expected value, both of which can be calculated from the Tor consensus file. Thus, we use $P'_{\text{Exit}}(j)$ to represent the expected probability for $F(j_{\text{Exit}} \mid i_{\text{Entry}})$'s approaching in our measurement period, the obtain of the other probability is analogous.

$$P'_{\text{Exit}}(j) = \frac{\text{Var}(P_{\text{Exit}}(j))}{E(P_{\text{Exit}}(j))} \tag{5}$$

Another challenge is that, due to the routing rules of Tor, the selection of a preceding node $i$ and our Middle node might slightly influence the candidate set for the subsequent node $j$, causing the actual probability of selecting node j as the Exit (denoted as $P^*_{\text{Exit}}(j)$ ) at a moment to be somewhat less than $P_{\text{Exit}}(j)$. However, such subtle difference between $P^*_{\text{Exit}}(j)$ and $P_{\text{Exit}}(j)$ is negligible comparing to the much more apparent and significant anomaly deviation awaiting detection between $F(j_{\text{Exit}} \mid i_{\text{Entry}})$ and $P'_{\text{Exit}}(j)$, thus such evaluation approach should not result in false positives.

In real Tor network, if users intentionally tend to fix certain Entry-Exit pairs during circuit construction, both $F(i_{\text{Entry}} \mid j_{\text{Exit}})$ and $F(j_{\text{Exit}} \mid i_{\text{Entry}})$ shall be simultaneously anomalous with significant deviation from their expected values. This indicates that when node $i$ is selected as the Entry, node $j$ is likely to be chosen as the Exit with an abnormal chance, and vice versa, suggesting that nodes $i$ and $j$ have an abnormal high dependency tendency during Entry-Exit pair selection.

We develop a statistical model to evaluate the frequency $F(j_{\text{Exit}} \mid i_{\text{Entry}})$ and $F(i_{\text{Entry}} \mid j_{\text{Exit}})$ of every selecting Entry node $i$ and Exit node $j$. We conduct a one-tailed binomial hypothesis [17] test to assess whether $F(j_{\text{Exit}} \mid i_{\text{Entry}})$ significantly deviates from the adjusted expected probability $P'_{\text{Exit}}(j)$. This test is well-suited for our analysis as it leverages exact probability calculations which are independent of sample size, providing a reliable method for identifying Usage anomalies. The null hypothesis posits that the theoretical selection probability of node $j$, $P'_{\text{Exit}}(j)$, does not significantly differ from the observed frequency $F(j_{\text{Exit}} \mid i_{\text{Entry}})$, while the alternative hypothesis suggests a significant deviation. The p-value, calculated by (6), represents the probability of observing at least $k$ successes out of $n$ trials. We reject the null hypothesis if p-value is less than the significance level $\alpha$, which is typically set to 0.05 or 0.001. In this study, to achieve more stringent anomaly detection results, we set the p-value threshold to 0.005, which in some researches [18][19] is considered a stricter significance level that can reduce false positives. If both $F(j_{\text{Exit}} \mid i_{\text{Entry}})$ and $F(i_{\text{Entry}} \mid j_{\text{Exit}})$ reject the null hypothesis, we strongly suspect anomalous usage between node pairs $i$ and $j$.

$$p = p(X \geq k) = \sum_{i=k}^{n} \binom{n}{i} P'_{\text{Exit}}(j)^i (1 - P'_{\text{Exit}}(j))^{n-i} \tag{6}$$

## 4.3 Clustering anomalous nodes

The proposed approach allow us to identify a large set of anomalous Tor nodes, we further cluster these node into different groups for exposing their association at the organization level. Intuitively, Tor nodes deployed by the same organization are expected to share certain common features. In this paper, we mainly consider three features as follows.

**Intrinsic Attribute Similarity**. This feature measures the similarity of some intrinsic attributes of Tor nodes. We observe that nodes hosted by the same Virtual Private Server (VPS) provider may adopt similar nicknames, allocate similar amounts of bandwidth, and run identical version of the Tor software. Furthermore, they may exhibit coordinated patterns of simultaneous online and offline activity. Therefore, we select five key attributes, including nickname characteristics, network attribute characteristics, geographical location, temporal behavior, and Tor version, to calculate a similarity score as the first feature. The considered attribute features are detailed in Appendix B. Base on these attributes, we train a machine learning model to output the probability of two anomalous nodes having similar behaviors as their similarity score. The training dataset is sourced from verified trusted node data, which is publicly available on the Tor Metric platform, comprising 2,256 positive samples and 2,256 negative samples. All samples are selected through random sampling to ensure fairness. We choose the best-performing model by testing and comparing several machine learning algorithms including the Naive Bayes classifier, Support Vector Machine (SVM), Logistic Regression (LR), and Random Forest. We evaluate their performance using metrics including accuracy, precision, recall and F1. Finally, we select the best model to calculate the intrinsic attributes similarity, i.e., $sim(a, b)$, of two nodes.

**Family Relationship**. Our second feature derives from the "family" configuration option in a Tor node. The "family" configuration information is defined by the node owner, explicitly indicating other nodes that belong to the same organization as this node. This can help users avoid selecting multiple nodes from the same organization when constructing circuits. Note that the "family" configuration in each Tor node is retrievable. We introduce $fam(a, b)$ in Equation (7) to indicate whether two nodes have been explicitly specified as in the same organization. Intuitively, anomalous nodes that explicitly declare their membership in the same family should be clustered together. Therefore, this feature can further improve clustering accuracy.

$$\text{fam}(a, b) = \begin{cases} 0, & a \text{ and } b \text{ in one family} \\ 1, & \text{otherwise} \end{cases} \tag{7}$$

**Abnormal Behavioral Pattern**. This feature describes the relationship of two nodes in term of their abnormal behavioral patterns, which can be identified through the method in Section 4.2. We introduce a connectivity metric $conn(a, b)$ in Equation (8) to quantify this relationship between nodes $a$ and $b$. Here, $N_c(a, b)$ represents the times of co-occurrence of nodes $a$ and $b$ in a same anomalous circuit.

$$\text{conn}(a, b) = e^{-N_c(a,b)} \tag{8}$$

To reduce the impact of substantial variances of $N_c(a, b)$, we apply a logarithmic function to mitigate the significant differences, thereby ensuring the feature more stable and reliable.

Finally, based on the above three features, i.e., $sim(a, b)$, $fam(a, b)$, and $conn(a, b)$, we employ a density-based clustering algorithm DB-SCAN [14] to effectively group anomalous Tor nodes exhibiting similar abnormal behaviors. For any two identified anomalous Tor nodes $a$ and $b$, the distance metric $D(a, b)$ is derived by

$$D(a, b) = \alpha \cdot sim(a, b) + \beta \cdot fam(a, b) + \gamma \cdot conn(a, b) \qquad (9)$$

To ensure the family group factor does not dominate the clustering process, we empirically set $\alpha$, $\beta$, and $\gamma$ to 1, 0.5, and 1, respectively.

## 5 EXPERIMENTAL EVALUATION

In this section, we conduct experiments on the real-world Tor network to evaluate our methods, and conclude several interesting observations about the organizational clustering of nodes involved in anomalous circuits.

### 5.1 Experiment setup

We conduct experiment on top of the real-world Tor network with more than 9,000 nodes [20] . We deploy 10 modified Middle nodes across various global regions to collect circuit data. These regions include New York, San Francisco, Amsterdam, Singapore, London, Frankfurt, Toronto, Bangalore, and Sydney. All nodes are deployed on a Digital Ocean VPS, running Ubuntu 20.04 (LTS) x64 with 1 GB of memory and 25 GB of disk space. Each node contains modified source codes for controlling relay functionality to record the timestamp, direction, Circuit Section ID, and the fingerprint of the next forwarding hop for every received or transmitted Cell. Over a 45-day period from March 1st to April 15th 2024, a total of 45,367,152 traces within circuit section records from these nodes are collected.

### 5.2 Node position identification results

We classify circuit sections of all collected traces according to the method in Section 4.1, resulting in 24 categories. With a classification accuracy of 100%, we can precisely discern Exit and prebuilt circuits for further validation. This process has also revealed some additional intriguing findings. For each of the 9 circuit types, we select the Entry-Middle section as a representative to estimate the proportions of different circuit types within the Tor network.

The pie charts in Figure 7 illustrate these distributions. Notably, prebuilt circuits account for 81% of all circuits, suggesting that a large majority are established but not used. Maintenance circuits make up 18%, supporting network operations like downloading consensus files or performing bandwidth tests. Only 1% of the circuits are actively user-initiated, with 18.5% classified as General circuits and 16.3% as Client-RPO circuits, suggesting that Tor users access both general and dark websites at comparable rates.

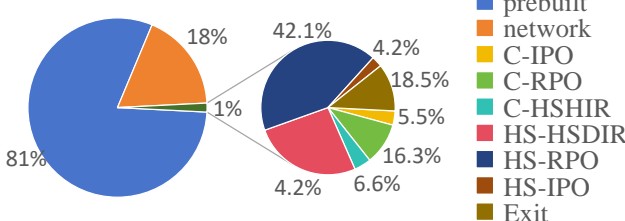

**Figure 7: Circuit type distribution**

### 5.3 Anomalous circuit detection results

We screen a total of 31,362 pairs of relays in the Exit circuit and 727,738 pairs of prebuilt circuits designated for Exit circuits, resulting in 759,100 pairs for anomaly detection. After deduplication, the final set of circuits to be examined includes 415,858 unique circuits.

Next, we conduct an anomaly detection on these circuits, and the results are presented in Table 1. We categorize Usage Anomalies into three cases. The first two represent abnormal selection probabilities for an Exit (or Entry) node when a specific Entry (or Exit) node is fixed. The third case represents node pairs exhibiting both types of anomalies simultaneously, which are classified as the final Usage Anomalies set. Based on a significance level of $p = 0.005$, the ratio of circuits where both the Exit probability given a fixed Entry and the Entry probability given a fixed Exit are anomalous should align with the expected value of 0.005. This type of anomaly is referred to as a Type I error [21], indicating that any hypothesis test will have approximately a 0.5% false positive rate. Notably, statistical results show that the ratio of circuits exhibiting both anomalies, compared to those where only the Exit probability given a fixed Entry is anomalous, reaches 11%, which is 22 times higher than the expected value. From a statistical perspective, this suggests that the majority of the identified anomalies are likely true positives, rather than Type I errors. Among the anomalous nodes, those found within the usage anomaly circuits are typically more dangerous. While these nodes may not explicitly violate Tor routing rules, they are more likely to be potential Sybils, posing a significant security threat to the network.

**Table 1: Statistics of circuit anomalies**

| Anomalies Type | Node Pairs | Number of Nodes |
|---|---|---|
| Routing | 4,198 | 1,641 |
| Usage(Entry fixed) | 6721 | 2549 |
| Usage(Exit fixed) | 2634 | 1979 |
| Usage(Both fixed) | 300 | 360 |

We present two representative pairs of anomalous nodes along with their configuration information in Table 2. Neither pair declares a family group identifier. Node pair 1 shows similar intrinsic attributes, suggesting they likely belong to the same organization. In contrast, node pair 2 has low intrinsic attribute similarity, with multiple differing attributes including a nearly one-month gap in First_seen_time. Despite low independent selection probabilities (approximately $10^{-5}$), this pair constantly appear together as an Entry-Exit pair in our experiment, suggesting strong coordination and potential organizational ties. **Notably, such an exposed anomalous pair by our approach cannot be detected solely by node configuration comparative methods.**

**Table 2: Anomalous node pairs with configuration**

| Config. | Node Pair 1 | | Node Pair 2 | |
|---|---|---|---|---|
| Fingerprint | ***4EC4 | ***E86C | ***7255 | ***67DC |
| Nickname | ysch*** | ysch*** | Andr*** | dee*** |
| Or Port | 9001 | 9001 | 9001 | 9001 |
| Adv. BW | 23375514 | 30798848 | 3145728 | 4166656 |
| Burst BW | 1073741824 | 1073741824 | 3145728 | 1073741824 |
| Obs. BW | 23375514 | 30798848 | 3395838 | 4166656 |
| Country | Germany | Germany | United States | United States |
| First Seen | 2023/8/2 18:00 | 2023/8/2 18:00 | 2023/8/9 5:00 | 2023/9/25 19:00 |
| Last Change | 2023/8/2 18:00 | 2023/8/2 18:00 | 2023/8/9 5:00 | 2023/9/25 19:00 |
| Tor Version | 0.4.8.9 | 0.4.8.9 | 0.4.8.8 | 0.4.7.13 |

## 5.4 Anomalous nodes clustering results

The performance of different machine learning models in node similarity calculation is summarized in Table 3. Among these models, SVM demonstrated the highest F1 score and is therefore selected as the final classification model.

**Table 3: Performance of different machine learning models in node similarity classification.**

| Method | Accuracy | Precision | Recall | F1 |
|---|---|---|---|---|
| Naive Bayes | 95.35% | 87.10% | 89.91% | 88.50% |
| **SVM** | **96.45%** | 88.55% | **92.32%** | **90.43%** |
| LR | 96.24% | **89.63%** | 90.10% | 89.87% |
| Random Forest | 95.33% | 88.77% | 89.00% | 88.88% |

We then perform clustering on all 1960 nodes involved in these abnormal circuits, aiming to uncover hidden organizational relationships. The results are shown in Figure 8, which presents the distribution of nodes based on their relational positions using MDS (Multi-dimensional Scaling) [22] for 2D visualization. DBSCAN (with eps = 0.15) identifies four dense clusters, each highlighted in distinct colors. The distribution of family groups across these clusters is summarized in Table 4. Cluster 1 and cluster 2 consist exclusively of nodes from Family Groups 1 and 2, indicating a strong user preference for selecting these groups as Entry-Exit pairs. In contrast, cluster 3 and cluster 4 contain members from multiple family groups along with discrete nodes. Importantly, these two clusters are gathered based on features other than strict family relationship. **Specifically, anomalous links of Discrete nodes in cluster 3 to Family Group 3 are exposed mainly due to their similar behavioral patterns, demonstrating the need to include the connectivity metric we proposed in assessing node associations.**

These observations suggest that there are hidden organizational relationships between family groups and discrete nodes, or even between different family groups. These organizations could potentially act as Sybils, launching attacks on the anonymity of the Tor network. To ensure the security of anonymous communication, Tor needs to implement stricter routing rules and conduct more thorough reviews of node organizations.

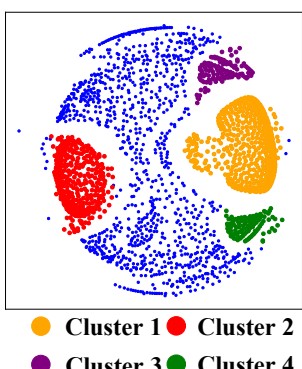

**Table 4: Family distribution for clusters**

| Clusters & Size | Family ID & Prop. |
|---|---|
| 1(420) | 1(100%) |
| 2(340) | 2(100%) |
| 3(113) | 3(75%), 7(2%), 17(2%), 22(5%), null(15%) |
| 4(102) | 4(82%), 8(2%), 34(3%), null(12%) |

● Cluster 1 ● Cluster 2
● Cluster 3 ● Cluster 4

**Figure 8: Potential malicious accomplices clustering result**

## 6 RELATED WORK

Some work studies on deanonymization attacks using traffic confirmation between pairs of compromised nodes. Biryukov [23] et al. use long-lived connections and differential scanning attacks to trace Tor users, highlighting vulnerabilities in user anonymity. Sun [24] et al. introduce routing attacks that exploit the asymmetric nature of Internet routing to deanonymize Tor users, demonstrating how Autonomous Systems can be leveraged to increase surveillance on the Tor network. In another work, Kwon [25] et al. conduct circuit fingerprinting attacks to passively deanonymize hidden services, showing the potential for identifying hidden service operators through passive monitoring techniques.

To prevent the attacks above, existed studies have explored the behavior and impact of malicious nodes within the Tor network. Jansen [26] et al. demonstrate how targeted resource exhaustion on Entry nodes can deanonymize and disable parts of the Tor network. Singh [27] et al. analyze the patterns and effects of Exit node blocking, revealing significant insights into systematic censorship of Exit nodes. Sanatinia [28] et al. investigate the extent and methods of malicious activities within HSDir nodes, emphasizing the need for robust detection and mitigation strategies. Additionally, Wang [29] et al. examine the vulnerabilities and exposure of bridge nodes, providing a comprehensive assessment of their susceptibility to discovery and blocking.

Some studies [25][29][30][26] attempt to classify specific circuits in the Tor network based on traffic characteristics. Compared to these works, our approach not only addresses the failures caused by updates in Tor (such as padding interference) but also provides a more detailed classification of the circuit positions of penetration nodes. Also, a similar method for identifying Sybil is proposed by Winter et al. [9]. In this study, the authors developed "sybilhunter," a system for detecting Sybil relays in the Tor network by analyzing their appearance such as configuration and behavior such as uptime sequences. In contrast, our work takes a deeper approach by examining the limitations of Tor's routing rules, revealing more complex Sybils from the perspective of node pairs in anomalous Tor circuits, which is ignored by previous methods.

## 7 CONCLUSION

This paper presents a novel approach to detecting anomalous circuits in the Tor network, uncovering potentially malicious nodes that pose significant risks to user anonymity. By leveraging data from controlled Middle nodes, we identify two primary types of anomalies: Routing anomalies and Usage anomalies. Using a comprehensive anomaly detection model, we successfully identified 1,960 anomalous nodes within the Tor network. Furthermore, through clustering analysis, we uncovered deeper organizational relationships among these nodes that are undetectable through simple configuration comparisons. Each of these organizations could potentially act as Sybils within the Tor network. Our findings suggest that certain nodes may deliberately collaborate to exploit vulnerabilities and bypass Tor's security measures. These results underscore the urgent need for enhanced routing policies and stricter security protocols to mitigate such risks, which is essential for safeguarding user privacy in everyday web activities.

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

# A   CIRCUIT SECTION TYPES

The following 9 circuit types with totally 24 types of circuit sections are classified in Section 4.1:

- **Exit Circuits:**
  - Entry-Middle
  - Middle-Exit
- **Client-HSDIR Circuits:**
  - Entry-Middle
  - Middle-Middle2
  - Middle2-HSDIR
- **Client-IPO Circuits:**
  - Entry-Middle
  - Middle-Middle2
  - Middle2-IPO
- **Client-RPO Circuits:**
  - Entry-Middle
  - Middle-HSDIR
- **HS-HSDIR Circuits:**
  - Entry-Middle
  - Middle-Middle2
  - Middle2-HSDIR
- **HS-IPO Circuits:**
  - Entry-Middle
  - Middle-Middle2
  - Middle2-IPO
- **HS-RPO Circuits:**
  - Entry-Middle
  - Middle-Middle2
  - Middle2-RPO
- **Network Status Circuits:**
  - Entry-Middle
  - Middle-Middle2
  - Middle2-relay
- **Prebuilt Circuits:**
  - Entry-Middle
  - Middle-Middle2

# B   SIMILARITY FEATURES

The following presents features for calculating the parameter $\text{sim}(a, b)$, which are used to facilitate clustering in Section 4.3.

## Nickname Features

- **nickname_length_difference**: The absolute difference in lengths of the node nicknames. Numeric value representing the difference in characters.
- **levenshtein_distance**: Levenshtein edit distance [31] between two nicknames. A numeric value representing the number of single-character edits (insertions, deletions, or substitutions) needed to change one nickname into the other.
- **longest_common_substring_length**: Length of the longest common substring found in the nicknames of two nodes. A numeric value representing the number of matching consecutive characters in the nicknames.
- **nickname_entropy_similarity_flag**: A binary flag (0 or 1). 1 if the Shannon entropy (complexity) of the two nicknames is similar (difference in entropy is less than 1), 0 otherwise.

- **nickname_pattern_similarity**: A binary indicator (0 or 1). 1 if the character pattern (lowercase, digits, uppercase) between the two nicknames is the same, 0 otherwise.

## Network Features

- **advertised_bandwidth_similarity**: A binary indicator (0 or 1). 1 if the difference in advertised bandwidth between the two nodes is less than 1000 bytes, 0 otherwise.
- **burst_bandwidth_similarity**: A binary indicator (0 or 1). 1 if the difference in burst bandwidth between the two nodes is less than 1000 bytes, 0 otherwise.
- **observed_bandwidth_similarity**: A binary indicator (0 or 1). 1 if the difference in observed bandwidth between the two nodes is less than 1000 bytes, 0 otherwise.

## Geographic Features

- **country_match**: A binary indicator (0 or 1). 1 if the nodes are located in the same country, 0 otherwise.

## Temporal Features

- **ip_port_change_recent**: A binary indicator (0 or 1). 1 if the time difference since the last IP or port configuration change is less than 0.1 hours, 0 otherwise.
- **first_seen_time_difference**: A binary indicator (0 or 1). 1 if the time difference between when the nodes first appeared on the network is less than 0.1 hours, 0 otherwise.
- **last_change_time_difference**: A binary indicator (0 or 1). 1 if the time difference between when the nodes last changed on the network is less than 0.1 hours, 0 otherwise.
- **uptime_sequence_distance**: Euclidean distance of the uptime sequence between two nodes over the past six months.

## Version Features

- **software_version_match**: A binary indicator (0 or 1). 1 if both nodes are running the same software version, 0 otherwise.