# OpenReview forum: "Do Not Trust What They Tell: Exposing Malicious Accomplices in Tor via Anomalous Circuit Detection"
_ACM.org/TheWebConf/2025/Conference — WWW 2025 Poster_

### Official Review · Reviewer_iU7f · 2024-11-19

**Novelty:** 5
**Technical Quality:** 5

**Review:**

The paper "Do Not Trust What They Tell: Exposing Malicious Accomplices in Tor via Anomalous Circuit Detection" tackles a significant challenge in the Tor network—identifying colluding, malicious nodes that perform deanonymization attacks by exploiting Tor's circuit construction vulnerabilities. The proposed solution, which combines traffic analysis, anomaly detection, and clustering techniques, offers a novel approach to uncovering hidden Sybil organizations within the Tor network by detecting anomalous Entry-Exit node pairs. This is a relevant and impactful contribution to enhancing privacy and security in anonymity networks.

Pros:


Comprehensive Methodology: The paper's methodology is well-structured, covering several stages: traffic data collection, node position classification, anomalous circuit detection, and clustering of malicious nodes. Each step is thoroughly explained, with formal statistical models used to detect anomalies in Entry-Exit pairings. The authors provide clear mathematical formulations for their models, making the work reproducible and scientifically rigorous.


Large-Scale Experimental Validation (Realistic evaluation): The experiments conducted on real-world Tor data, involving over 45 million circuit traces, lend significant weight to the paper's claims. The deployment of 10 modified Middle nodes in various global locations over a 45-day period demonstrates the practicality of the approach. The paper also presents thorough results, identifying 1,960 anomalous nodes, clustering them into potential malicious organizations, and providing meaningful insights into the structure and behavior of Sybil groups within Tor.

Security Implications and Practicality: The work has direct and significant implications for improving the security of Tor. By revealing hidden organizational relationships between malicious nodes, the paper offers actionable insights that can inform future improvements in Tor's routing protocols. Furthermore, the approach is designed to be practical, with minimal false positives, as evidenced by the statistical rigor behind the anomaly detection model.

Cons:

Limited Scope of Anomalous Behavior Detection: While the paper focuses on detecting anomalous Entry-Exit node pairings, it does not fully explore other types of potential attacks within Tor, such as traffic analysis between Middle nodes or bridge node vulnerabilities. The focus on Entry-Exit pairs, although effective, leaves out other critical areas of vulnerability that could benefit from similar anomaly detection techniques.

Complexity of Implementation: The solution requires the deployment of modified Middle nodes, which may not always be feasible in practice. Additionally, the requirement for a large-scale deployment across different regions could be resource-intensive. While the approach is well-suited for research or targeted deployment, it may face challenges in large-scale real-world implementation, especially considering Tor’s reliance on volunteer-operated nodes.

Lack of Real-Time Performance Evaluation: The paper does not provide a detailed evaluation of the real-time performance of the proposed system. While the data collection and analysis processes are well-explained, the potential latency introduced by deploying Middle nodes, analyzing traffic, and detecting anomalies in real-time circuits is not fully discussed. This is particularly relevant in operational environments where low-latency communication is critical.

Potential for Evasion by Adversaries: The system’s reliance on detecting anomalous behavior through traffic patterns and frequency deviations might be evaded by sophisticated attackers. For example, Sybil nodes could adjust their behavior to avoid detection by mimicking normal traffic patterns. While the paper addresses explicit and implicit node associations, a more thorough analysis of how adversaries might adapt to evade detection would strengthen the work.

**Questions:**

Could the authors discuss the potential challenges in deploying this system at scale in a fully decentralized network like Tor, particularly in terms of the resources required to maintain multiple Middle nodes across various regions?

How does the proposed system perform in real-time scenarios? Is there a potential latency impact on users' connections when Middle nodes are actively collecting and analyzing traffic?

What strategies could be used to counter adversaries that may attempt to modify their behavior to avoid detection? How resilient is the system to adaptive or evasive tactics by Sybil attackers?

**Reviewer Confidence:**

3: The reviewer is confident but not certain that the evaluation is correct

**Scope:**

4: The work is relevant to the Web and to the track, and is of broad interest to the community

---

### Official Review · Reviewer_Nz9C · 2024-11-24

**Novelty:** 4
**Technical Quality:** 3

**Review:**

The paper proposes a novel method for detecting anomalous circuits in the Tor network, focusing on two anomaly types: routing anomalies and usage anomalies. Using strategically deployed modified middle nodes, the authors collect traffic data and classify circuits to identify anomalies. Their approach uncovered 1,960 anomalous nodes, and subsequent clustering analysis revealed potential hidden organizations threatening network security.

pros:
- The study shifts focus to usage anomalies, a relatively unexplored area in Tor anomaly detection, complementing previous work on routing anomalies and family relationships.
- The method is interpretable, and effective, making it practical for real-world application.

cons:
- While the study makes significant contributions, it raises ethical concerns due to the use of modified middle nodes for traffic data collection, potentially conflicting with Tor's privacy principles.
- The experimental setup lacks sufficient detail and baseline comparisons, limiting the reproducibility and comprehensive evaluation of the proposed method.

**Questions:**

- The proposed method relies on the deployment of self-built intermediate nodes, which, strictly speaking, involves introducing self-built nodes. Could this introduce additional security risks, and does it potentially conflict with the paper’s stated objective of protecting user privacy?
- Section 4.1 mentions achieving 100% accuracy with a decision tree for circuit classification. Could the authors elaborate on how the 200 services were configured, whether the data adequately simulates real-world Tor traffic, and what recall or other performance metrics were observed?
- For the anomalous circuit detection, how to evaluate the results?
- Section 5.3 does not include sufficient comparisons with prior works, such as [9]. How does the proposed method perform in comparison to existing approaches?
- In Section 5.4, clustering analysis reveals potential Family relationships among anomalous nodes, but the discussion of these results is limited. Could the authors expand on how these relationships could be validated or associated with specific entities or organizations, and what insights can be drawn regarding their intent or behavior?
- The abstract mentions the necessity of mitigation strategies to address the security risks posed by anomalous circuits, but the paper does not provide concrete details. What specific mitigation strategies do the authors propose, and how might these be implemented in the Tor network to reduce risks?
- Upon reviewing Figure 8, which presents the clustering results of the nodes based on their relational positions, it has been noted that there is a lack of clarity regarding the representation of the blue points within the visualization. This omission is significant as it affects the reader's ability to fully understand and interpret the clustering outcomes.

**Reviewer Confidence:**

3: The reviewer is confident but not certain that the evaluation is correct

**Scope:**

4: The work is relevant to the Web and to the track, and is of broad interest to the community

---

### Official Review · Reviewer_cN9W · 2024-11-27

**Novelty:** 3
**Technical Quality:** 4

**Review:**

Thanks for submitting your work to WWW 2025. I appreciate the authors' efforts to improve the anonymity of the Tor network through detecting potentially colluding relays (entry-exit pairs).

**Summary**
This papers targets the long-existing security task in the Tor network, namely, detecting anomalous circuits and colluding Tor relays. To achieve it, the authors proposed and implemented a three-step detection method: 1) classifying types of circuits and circuit sections; 2) identifying anomalous pairs of entry-exit nodes in exit or pre-built circuits; 3) clustering anomalous nodes into the underlying organizations or sybil campaigns.


**Pros in Short**
- It targets an import security task, namely, detecting anomalous circuits and nodes in the Tor network.
- The proposed detection methodology is promising.

**Cons in Short**
- The rationale underpinning the proposed detection methodology is missing.
- The implementation and evaluation of the detection methodology is not solid.
- Most observations are not well supported, and some claims regarding the maliciousness of the anomalous nodes should be toned down.

My first concern resides in the definition of **anomalous** circuits/nodes as well as the connection between anomalous circuits/nodes and colluding nodes used in de-anonymization attacks.  This study targets the detection of routing anomalies and usage anomalies. However, it deserves more discussion on whether a circuit violating the Tor routing recommendations should be considered as a routing anomaly, and whether entry-exit node pairs of higher usage volume should be considered as anomalous. Particularly, a client may use the same entry-exit pair to send a large volume of traffic flows, depending on how the authors counted the traffic traces, such a pair can be over-counted and then be considered as anomalous.  Also, even such cases can be considered as anomalies, no concrete evidences are there to support that they are intended for traffic de-anonymization attacks. Without such evidences, the security value of detecting such seemingly anomalous circuits is very limited.

My second concern resides in the solidness of the proposed methodology.  First, regarding the classification of circuit types, the authors didn't clarify whether the groundtruth was collected with diverse variants of Tor clients and hidden services, which is critical to decide the representativeness of the groundtruth. Then, regarding clustering anomalous nodes, some features to compute similarity (e.g., nicename bandwidth, the Tor version) can be easily evaded by true attackers, and the adoption of such features undermine the effectiveness of the resulting clustering algorithm. Also, to build up a machine learning model for similarity comparison, a training data was adopted from the Tor Metric Platform. However, no reference was given for such a dataset, and necessary description of this data is also missing.  It is thus unclear why such a dataset is sufficient to train a robust model that can accurately compare similarity between a pair of anomalous nodes.

My third concern is regarding what concrete and deterministic observations we can distill from this study. Applying the detection methodology to real Tor traffic, the authors distilled some security observations in Section 5. Across these observations, the authors tried to connect anomalous nodes/clusters to sybil attacks, however, the authors failed to provide sufficient evidence to support such claims. I recommend the authors to double check these observations and tone down statements that are not well supported with evidences.

Lastly, there are statements that are ambiguous or even misleading.
1) In Section 3.1, the authors mentioned that "trusted users who donate nodes to the attacker". What is the definition of trusted users? If they were benign users, why do they donate nodes to attackers?
2) In some statements and figures (e.g., Figure 4), the authors used sybils and anomalous circuits equivalently. However, they are different things. Anomalous circuits and nodes are ones obviating the regular usage/routing patterns, which however is not sufficient to pin them on the board of sybils.

**Questions:**

- What  deterministic  and well-supported security observations we can distill from this study?
- Can you provide more details about the dataset for comparing node similarity?
- Can you share the plan for data/code release?

**Reviewer Confidence:**

3: The reviewer is confident but not certain that the evaluation is correct

**Scope:**

4: The work is relevant to the Web and to the track, and is of broad interest to the community

---

### Official Review · Reviewer_kPSC · 2024-11-29

**Novelty:** 4
**Technical Quality:** 4

**Review:**

The paper introduces a pioneering circuit classification method based on anomaly traffic detection. By capturing and analysing traffic at relay nodes, it effectively identifies anomalous circuits and potential malicious nodes within the Tor network, demonstrating significant technical innovation. The study employs modified relay nodes deployed globally to collect large-scale real-world data. The method’s effectiveness is robustly validated using statistical models and machine learning algorithms, with clear and persuasive results. In addition, by using density-based clustering algorithms, the research uncovers potential links between anomalous nodes, revealing possible hidden organisations and collaborative behaviours. This provides a deeper understanding of Sybil attacks.

However, the method relies on deploying custom relay nodes to capture user traffic. Although intended for research purposes, this approach could increase user privacy exposure and may conflict with the privacy ethos of the Tor network.
The probabilistic models based on temporal dynamics, while capable of detecting anomalous correlations, may be sensitive to network state changes and sampling time windows, potentially affecting the model’s stability and robustness.
The experiments involve relay node deployment at only ten global locations. This geographical and numerical constraint may limit the detection capability for anomalous nodes in certain regions, impacting the comprehensiveness of the findings.

**Questions:**

When deploying modified relay nodes, what measures were taken to minimise potential privacy risks to regular users? Are there alternative, more privacy-preserving data collection methods under consideration?

Can the statistical model maintain consistent detection results across different periods and varying network states, especially given the fluctuations in node selection probabilities?

As the method scales to larger network environments or more complex concealed node behaviours, will its performance and accuracy degrade significantly? Are there plans for further optimisation to address such challenges?

**Reviewer Confidence:**

3: The reviewer is confident but not certain that the evaluation is correct

**Scope:**

4: The work is relevant to the Web and to the track, and is of broad interest to the community

---

### Official Review · Reviewer_hRVE · 2024-11-30

**Novelty:** 4
**Technical Quality:** 3

**Review:**

This paper detects anomalous circuit using machine learning. The authors deploy their own Middle Relay nodes in the real-world Tor network to collect the real-world circuit data to test their detection. The paper however has some deficiencies and challenges.

The detection targets are the anomalies caused by the Tor client (whose anonymity Tor is protecting). More specifically, the Routing anomalies are caused by the Tor client over-ruling the Tor’s guideline protocol to manually select the relay nodes; the Usage anomalies also seem to be from the Tor client’s malpractice, as opposed to choosing them randomly which will make the frequency/occurrence of the chosen pairs to be more even. This is different from the typical security defense research where the anomalies are caused by the attacker and not the victim self’s malpractice. In this case, it seems that the Tor client (whose anonymity is being attacked) can follow the Tor-recommended protocol or the best practice (such as random selection) to eliminate the anomalies.

It is unclear who the attackers are in the evaluation and how they are reflected in the evaluation results.

The related work section does not clarify the research gaps, as the comparisons are handwavy and ambiguous. It is unclear if the research contributions beyond [25,29,30,26,9] are significant or marginal/derivative (e.g., implementation and update-based differences). Also, there seems to be inconsistency between Introduction and Related Work sections. While the Introduction states that this is the first approach for detecting anomalous circuits, the related work section treats and contrasts with some related works which have the same goal.

Section 2.1 introduces HS, HSDIR, etc. However, these background are not much used in the research presentation and thus have limited value.

**Questions:**

Can the anomalies being detected be fixed/omitted if the Tor client follows the Tor protocol and recommendations, more specially, for how to select the relay nodes to construct the circuit?

How to know which relay nodes are the attackers? Also, are the attacker identifications based on speculations or evidence?

For the accuracy results, what are the ground truths and how are they derived?

What are the concrete and precise differences from the related work, and why are those differences significant and important?

Does Tor have some measures against Sybil? Are there biases in relay selections? How are the consensus weights determined? How are these relevant (or not relevant) to Sybil threat and this research work?

**Reviewer Confidence:**

3: The reviewer is confident but not certain that the evaluation is correct

**Scope:**

4: The work is relevant to the Web and to the track, and is of broad interest to the community